# Knowledge management process, knowledge based innovation: Does academic researcher's productivity mediate during the pandemic of covid-19?

**Fazal ur Rehman**[1]*, **Hishamuddin Ismail**[2], **Basheer M. Al Ghazali**[3], **Muhammad Mujtaba Asad**[4], **Muhammad Saeed Shahbaz**[5], **Ali Zeb**[6]

1 University of Lakki Marwat, Lakki Marwat, Pakistan, 2 Multimedia University Malaysia, Cyberjaya, Malaysia, 3 King Fahd University of Petroleum and Minerals, Dhahran, Saudi Arabia, 4 IBA Sukkar University, Sukkur, Pakistan, 5 SZABIST, Karachi, Pakistan, 6 UTHM, Parit Raja, Malaysia

* fazal_marwatpk@yahoo.com

**Data Availability Statement:** All data are fully available in the uploaded Supporting Information files without restrictions.

## Abstract

Drucker's knowledge-worker productivity theory and knowledge-based view of the firm theory are widely employed in many disciplines but there is little application of these theories in knowledge-based innovation among academic researchers. Therefore, this study intends to evaluate the effects of the knowledge management process on knowledge-based innovation alongside with mediating role of Malaysian academic researchers' productivity during the Pandemic of COVID-19. Using a random sampling technique, data was collected from 382 academic researchers. Questionnaires were self-administered and data was analyzed via Smart PLS-SEM. Knowledge management process and knowledge workers' productivity have a positive and significant relationship with the knowledge-based innovation among academic researchers during the Pandemic of COVID-19. In addition, knowledge workers' productivity mediates the relationship between the knowledge management process (knowledge creation, knowledge acquisition, knowledge sharing, and knowledge utilization) and knowledge-based innovation during the Pandemic of COVID-19. Results have also directed knowledge sharing as the key factor in knowledge-based innovation and a stimulating task for management discipline around the world during the Pandemic of COVID-19. This study provides interesting insights on Malaysian academic researchers' productivity by evaluating the effects of knowledge creation, acquisition, sharing, and application on the knowledge-based innovation among academic researchers during the Pandemic of COVID-19. These useful insights would enable policymakers to develop more influential educational strategies. By assimilating the literature of defined variables, the main contribution of this study is the evaluation of knowledge creation, acquisition, sharing, and utilization into knowledge-based innovation alongside the mediating role of knowledge workers productivity in the higher education sector of Malaysia during the Pandemic of COVID-19.

**Funding:** The authors received no specific funding for this work.

**Competing interests:** The authors have declared that no competing interests exist.

## Introduction

The far-reaching innovation has changed the process and capacity of production in many fields of endeavor around the globe. This rapid innovation has also created challenges for policymakers, management professionals, researchers, and practitioners in enhancing the productivity of workers during the Pandemic of COVID-19. Peter Drucker, who is called the founder of modern management theory, has highlighted the issue as an important factor for increasing the output of workers. As production is often the prime objective in many organizations and more often management is oriented towards that. Therefore, scientific management is one of the key techniques needed to enhance the level of productivity of academic researchers as regards task efficiency and research publications [1–6]. In the modern scientific era, the service sector is predominantly driven by knowledge-based techniques and the digital economy to provide better quality services and enhance the level of productivity during the Pandemic of COVID-19. Therefore, the prime challenge for management practitioners is to increase the capacity of productivity of knowledgeable workers, solving problems and completing tasks especially in the academic sector during the Pandemic of COVID-19. [3, 5–8]. As noted earlier, management practitioners and strategists widely focusing on the need to increase the productivity of academic researchers. They also emphasized evaluating the impacts of the knowledge management process on individuals with concerns about the soft and hard aspects of tasks as recommended in the literature [9]. In such circumstances, evaluating the effects of the knowledge management process on the knowledge-workers productivity among academic researchers (individual workers) could be a novel contribution in the Malaysian academic environment [3–6, 10], which could attract literature attention towards this part of the world during the Pandemic of COVID-19. To this end, the problems in this context are presented in the following passages.

Firstly, the ultimate effect of knowledge management is to get innovation for competitive advantage [11]. Thus, the interesting consequence of knowledge management is the knowledge management process desirable for implementing its architecture in organizations [12]. In such arena, studies have noted the association between knowledge management processes, practices, and infrastructure and the consequences of innovation [13–17]. However, less focus had been given on the evaluation of the effects of the knowledge management process on knowledge-based innovation among academic researchers in Malaysia. Secondly, the impacts of knowledge management process on the employee's productivity with regards to task efficiency have been tested in the IT sector [18–20] but less focus was made to evaluate the effects of knowledge management process on academic researcher's productivity in Malaysia during the Pandemic of COVID-19. Consequently, this study has noted a gap of knowledge to evaluate the effects of knowledge management process on knowledge-based innovation along with the mediating role of knowledge workers productivity based on Drucker's knowledge-worker productivity theory and knowledge-based view of the firm theory among Malaysian academic researchers during the Pandemic of COVID-19. The issue of Pandemic of COVID-19 has widely affected the learning environment, students and teachers interaction, discussion sessions, participation in seminars and conferences to generate new ideas, and academic researcher's productivity in Malaysian educational institutions. Malaysian institutions regularly organize knowledge-sharing sessions to promote the research culture where researchers express their opinions, discuss new research ideas, and debate current research trends. However, the problem of Pandemic COVID-19 has widely affected these knowledge-sharing events.

Hence, this study is theoretically approaching the knowledge gap by finding the answers to the following raised research questions. Firstly, to what degree do knowledge creation,

knowledge acquisition, knowledge sharing, and knowledge utilization affect the knowledge-based innovation among Malaysian researchers during the Pandemic of COVID-19? Secondly, to what degree do knowledge creation, knowledge acquisition, knowledge sharing, and knowledge utilization affect the academic researcher's productivity during the Pandemic of COVID-19? Thirdly, to what extent does knowledge workers' productivity affect the knowledge-based innovation among Malaysian academic researchers during the Pandemic of COVID-19? Fourthly, does knowledge workers' productivity mediate the relationship between knowledge creation, knowledge acquisition, knowledge sharing, and knowledge utilization with the knowledge-based innovation among Malaysian academic researchers during the Pandemic of COVID-19? So, to address these questions, this study has proposed that the outcome of knowledge management is innovation and could cultivate the knowledge workers' productivity which leads to the conclusion that Drucker's knowledge workers productivity theory outcome (knowledge-worker productivity) should be treated as innovation. Hence, the prime objective of this study is to examine the mediating role of knowledge-workers productivity in the relationship between knowledge creation, knowledge acquisition, knowledge sharing, and knowledge utilization, and knowledge-based innovation among Malaysian academic researchers during the Pandemic of COVID-19.

To achieve these objectives, the authors needed to conduct a study in the higher education sector of Malaysia to clarify the answers to the above questions. Hence, this study contributes to relevant literature by evaluating the effects of the knowledge management process on the knowledge-based innovation among Malaysian academic researchers during the Pandemic of COVID-19. It also examines the effects of knowledge workers' productivity on knowledge-based innovation among Malaysian academic researchers during the Pandemic of COVID-19. This study contributes to the body of knowledge by assessing the mediating role of knowledge workers' productivity in the relationship between knowledge creation, knowledge acquisition, knowledge sharing, knowledge utilization, and knowledge-based innovation during the Pandemic of COVID-19. This study further explored the application of Drucker's knowledge-worker productivity theory and knowledge-based view of the firm theory in the development of a theoretical framework. This study was able to merge literature of knowledge management process, knowledge workers productivity, and knowledge-based innovation, capitalizing on the research findings from emerging economies in validating the model during the Pandemic of COVID-19. The application of PLS-SEM in the proposed research model is also a novel contribution. Therefore, the structure of this study starts with an introduction followed by theoretical background to explain the conceptions of the defined constructs alongside the conceptual framework. The methodology followed as third in the structure, while results are plainly explained in the fourth section of this study. Discussion, implications, and conclusion are presented in the last part of this article.

## Theoretical background

The foundation of this study is based on Drucker's knowledge-worker productivity theory and knowledge-based view of the firm theory. Drucker's knowledge-worker productivity theory highlights six factors of knowledge-worker productivity [3, 6, 21, 22] as explained in the following; possessing capabilities to understand and solve various tasks, knowledge-workers work should be knowledge-oriented, possess autonomy, innovative, keep continuity in learning, focus on creating quality and quantity, and knowledge-workers should be treated as valuable and intellectual assets, not as a cost of management. Whereas knowledge-based view of the firm theory classifies a firm into mix heterogeneous knowledge assets to gain competitive advantage at the firm level [23], which demands valuable, non-imitable, and rare knowledge of

a firm [24], that follows the process of knowledge creation, acquisition, knowledge transfer, and knowledge application [8, 25, 26].

## Innovation

Innovation is the invention, creation, development, and application of new ideas, solutions to problems, production of new products and services to improve the image and efficiency of an organization for customer satisfaction [27]. It has been classified as traditional innovation and knowledge-based innovation. Where, traditional innovation is the production of new products to improve customer problem-solving processes and satisfy the dynamic needs of all stakeholders [28]. The study has clarified the constructs in the context of product and the problem-solving innovation for consumers. While knowledge-based innovation only contributes the knowledge creation and application due to its pioneer characteristics which result in the production of new products and processes. Hence, knowledge-based innovation is the construction of new ideas, creation, and application of new knowledge in the context of novel products and services to solve customers' problems [12]. However, this study has focused on product innovation, process innovation, problem-solving innovation, service innovation, and radical innovation to evaluate the academic researcher's approach towards greater productivity.

**Product innovation.** Product innovation is the introduction of new products into the market or modification of the existing products in terms of certain features, like quality, packaging, flavor, functions [29], and the organizational learning process injected to improve the innovation capabilities and to solve the problems in order to gain an advantage over competitors [30]. It is a new process or method used to solve problems in a better way or offering a new product that addresses the market requirement based on stakeholders' expectations. Product innovation reduces cost and time of production, enhances efficiency, improves quality, and provides new opportunities for effective use of resources.

**Process innovation.** Process innovation has been attracted very little attention in the literature in the context of knowledge and extensively perceived intimately related to product innovation [31]. Widely, it is the new elements introduced by an organization's to produce a product or render a service with the aim of achieving lower costs and/or higher product quality. However, in the present arena, the learning methods and techniques, and the application of new technology to increase the academician's productivity are the components of process innovation. In addition, the application of active learning techniques in developing new ideas and concepts among academicians is perceived as knowledge-based process innovation. "Process innovation (aligning resources and capabilities) improves the learning system by improving technologies, products, and processes and by reducing or eliminating redundancies and problems" [29].

**Service innovation.** Service innovation is perceived extensively as part of employment in literature rather than manufacturing, isthe solution of the problem and the intangible combination of skills and processes, and includes informative knowledge-based services, health, education, and others. Concisely, it is the strategy explored to deliver intangible services by ensuring effective integration among researchers for greater learning outcomes. Service innovation focuses on the specific and systematic innovation used in structuring a system for solid, unique, and efficient services to increase productivity. Therefore, the trend of innovative services has emerged as a technique for sustainable growth, productivity, and learning outcome. Service innovation is the development of entirely new services or modifications in the existing services in accordance with the requirements [29].

**Radical innovation.** It is the exploration of higher uncertainties and "change that sweeps away much of a firm's existing investments in technical skills and knowledge, designs,

production technique, plant and equipment" [31]. It focuses on the incremental acquisition of new knowledge and consequences to enhance performance. Radical innovation transforms the structure of the new domains of learning that creates improvement in productivity. It emphasizes shifting from existing knowledge towards a novel and unique invention in the context of future innovation [32]. It exclusively emphasized the need for technological, social, and cultural innovations in building an advanced productive society.

**Problem solving capabilities.** It is the ability to realize the existence of problems, wisely analyze the situation, searching for alternatives solutions, solving the problems with intelligent solutions, and evaluating the meaningful results. The problem-solving capabilities are anticipations of the occurrence of problems, preventing through various barriers and mitigating its effects for fruitful outcomes. Problem-solving capability is a technical intelligence used to handle and analyze problematic situations in order to produce innovative and learnable outcomes.

## Knowledge-worker productivity

It is the ratio of production units [2, 5, 6, 33], which has been divided into traditional and knowledge-based productivity. Where traditional productivity is concerned with the production of manual workers and knowledge-based productivity is concerned with the enhancement of productivity of knowledge workers. In the 20th century, firms were generally regarded as production-oriented [4, 5, 6, 22]. As Peter Drucker has posited that "the productivity of manual workers was the striking challenge for management practitioners in the 20th century" [6, 7, 22]. While in the 21st century, firms are mostly focusing on enhancing the productivity of knowledge workers, turning into service and quality-oriented, highly innovative as regards customization and modification of their products and services [5, 34]. In the current age, knowledge-workers are mostly assigned with unstructured and intellectual responsibilities as compared to manual tasks [6, 3, 22]. Therefore, Peter Drucker had clarified that the most important task for management practitioners is to enhance the productivity of knowledge workers in the 21st century [3]. Furthermore, in clarifying the concept of knowledge-workers productivity, it is imperative to differentiate between knowledge work and knowledge workers [21]. Where knowledge work is the type of cognitive and intellectual work in a practical manner for producing and employing new knowledge [3, 7, 35]. While, the knowledge worker is that individual who utilizes and creates knowledge to generate further innovative knowledge for the development of new products and services [7, 22, 35]. In such circumstances, academic researchers can be considered knowledge workers [3, 4]. In such an arena, this burning issue has reinforced the need to collect data from academic researchers in the Malaysian universities for the conduct of this study during the Pandemic of COVID-19.

## Knowledge management

Knowledge Management is an interesting role and function of an organization to create, learn, enhance, share, organize and utilize knowledge in order to improve the level of innovation and efficiency in the overall performance of the organization [11, 36, 37]. Literature has highlighted two main parts of knowledge management, which are knowledge management environment and knowledge management processes. Though, the knowledge management process is treated as an important component of knowledge creation and enduring organizational support in improving knowledge management[11]. Henceforward, the primary focus of this study is tailored towards knowledge management processes rather than the knowledge management environment. The knowledge management process is the acquisition, sharing and transferring, creation, using and maintaining, and application of Knowledge[12]. Precisely, it is the process of knowledge creation, sharing, and utilization [8, 38], and these factors are treated as the

main processes of knowledge management [11]. In addition, codification and personalization are the two prime strategies for the flow of knowledge [8], where codification is related to the extraction and storage of knowledge and personalization is concerned with the direct interactions of humans to share knowledge [8, 24]. Thus, four main components of the knowledge management process are explained in detail in the following sections:

## Knowledge creation

Knowledge creation is the continuous process of creating new information based on the existing practice of organizational knowledge conception theory [11, 39], such as socialization, combination, externalization, and internationalization [16, 39]. The conceptions of these processes can provide the opportunity, motivation, capability, and perceived importance to create knowledge [11, 40].

**Knowledge acquisition.**   John Locke has introduced the concept of knowledge acquisition while describing the human mind at the birth time[41, 42], they further argued that knowledge is only acquired with experience [43]. Knowledge acquisition is the collection of information to build opinions and increase understanding of problems solving [44]. In addition, "knowledge acquisition includes the elicitation, collection, analysis, modeling, and validation of knowledge" [45]. Precisely, knowledge acquisition starts with the birth of a human who is further developed through social interactions and experience.

**Knowledge sharing.**   Knowledge sharing is the dispersion, donation, and collection of useful knowledgeable information among the different units of a firm [46], where workers transfer their views among colleagues to enhance the level of understanding [47]. Formal and informal donation and collection are the different sorts of sharing knowledge [47, 48]. Concisely, knowledge sharing is an interesting mechanism for enhancing knowledge and understanding employees as compared to knowledge generation [14]. Intrinsic and extrinsic incentives, social and organizational motivation, values and benefits, support and appreciation from the top leadership are the fundamental features of knowledge sharing [49].

**Knowledge utilization.**   Knowledge utilization is the practical application of knowledge in accomplishing different organizational tasks [50]. It is the application of knowledge in the completion of tasks that have been shared in an organization [51], and becomes a part of organizational processes in problem-solving through integration [15]. Knowledge utilization is one of the most important parts of knowledge management processes as compared to knowledge creation and sharing owing to its practical application [52].

**Relationship between knowledge management process and knowledge-based innovation.**   The theoretical and empirical literature had appraised the organizational approaches that effectively managed their knowledge assets perform better than those who do not manage it [53, 54], as there is a universal relationship between knowledge management process and innovation. In the same vein, [11] found a positive relationship between the knowledge management process and the level of innovation. More accurately, other studies established a positive relationship between knowledge creation and innovation [11, 15–17, 55, 56]. Similarly, [57] has indicated a positive relationship between knowledge acquisition and innovation. While, several authors have found a positive relationship between knowledge sharing and innovation [11, 13–15, 48, 50, 54, 58–60]. Equally, some studies have observed a positive relationship between knowledge utilization and innovation [15, 50, 51, 61, 62]. Arguably, the comprehensive review done on previous literature has opened a new chapter of research in the field of knowledge management and innovation. Consequently, this study developed the following hypotheses;

*H1*: *There is a positive and significant relationship between knowledge creation and knowledge-based innovation among Malaysian researchers during the Pandemic of COVID-19.*

*H2*: *There is a positive and significant relationship between knowledge acquisition and knowledge-based innovation among Malaysian researchers during the Pandemic of COVID-19*

*H3*: *There is a positive and significant relationship between knowledge sharing and knowledge-based innovation among Malaysian researchers during the Pandemic of COVID-19*

*H4*: *There is a positive and significant relationship between knowledge utilization and knowledge-based innovation among Malaysian researchers during the Pandemic of COVID-19*

**Relationship between knowledge management process and knowledge-worker productivity.** In the context of the knowledge-based view of the firm theory, the approach for effectively managing knowledge resources could enhance the level of innovation [11, 63], to which innovation becomes the result of knowledge-workers productivity. Therefore, it can be said that there is a relationship between the knowledge management process and knowledge-worker productivity [20]. Likewise, Drucker's knowledge workers productivity theory expatiate on; understanding of the task, giving autonomy, continuous innovation, focus on quality, learning and teaching by sharing knowledge and treating the knowledge-workers as valuable assets when it comes to their productivity [1]. In the same view, the knowledge management process was found to have a relationship with these factors [64, 65]. In addition, studies have noted a relationship between knowledge management and knowledge-workers productivity [18–20]. On the other hand, [20] have diverted their focus and observed a positive relationship between the knowledge management process and task efficiency. Hasas and Hansen [19] posited that knowledge sharing is a motivational factor that enhances quality and competencies at the workplace. Similarly, Ali [66] directed his focus towards the financial sector and noted that knowledge-sharing practices have a positive impact on work efficiency, competencies, performance, and customer satisfaction. Therefore, it can be concluded that knowledge creation leads to improvement in production [11], and reduces mistakes [14, 47]. Whereas the utilization of knowledge could lead to an effective problems solving capability[50]. Based on the findings from previous studies, this research hypothesized that;

*H5 There is a positive and significant relationship between knowledge creation and knowledge workers productivity among Malaysian researchers during the Pandemic of COVID-19*

*H6*: *There is a positive and significant relationship between knowledge acquisition and knowledge workers productivity among Malaysian researchers during the Pandemic of COVID-19*

*H7*: *There is a positive and significant relationship between knowledge sharing and knowledge workers productivity among Malaysian researchers during the Pandemic of COVID-19*

*H8*: *There is a positive and significant relationship between knowledge utilization and knowledge workers productivity among Malaysian researchers during the Pandemic of COVID-19*

**Relationship between knowledge-worker productivity and knowledge-based innovation.** In the context of the knowledge-based view theory of the firm, an improvement in the development of human knowledge can enhance innovation and performance in an organization [9, 63]. While in the views of Drucker's knowledge-worker productivity theory, the effectiveness of knowledge-workers enhances the level of firm innovation [1]. Furthermore, Ramezan [67] argued that based on empirical research design, knowledge creation at the

individual level, knowledge transfer, knowledge utilization, knowledge workers' productivity are related to innovative behavior in an organization. Additionally, Butt *et al*. [68] found that knowledge workers' productivity partially mediates the relationship between individual knowledge management engagement and innovation. Contrarily, customer feedback and workers' self-reflection were found to have a positive effect on innovation processes and outcomes. Such a relationship was due to mutual knowledge of the novelty and difficulty of job tasks. The coordination of self-reflection and customers' feedback can improve innovation processes and outcomes by creating novel solutions and products for the consumers [1, 50, 69]. This study, therefore, hypothesized that;

*H9*: *There is a positive and significant relationship between knowledge workers productivity and knowledge-based innovation among Malaysian researchers during the Pandemic of COVID-19*

*H10*: *Knowledge-worker productivity mediates the relationship between knowledge creation and knowledge-based innovation among Malaysian researchers during the Pandemic of COVID-19*

*H11*: *Knowledge-worker productivity mediates the relationship between knowledge acquisition and knowledge-based innovation among Malaysian researchers during the Pandemic of COVID-19*

*H12*: *Knowledge-worker productivity mediates the relationship between knowledge sharing and knowledge-based innovation among Malaysian researchers during the Pandemic of COVID-19*

*H13*: *Knowledge-worker productivity mediates the relationship between knowledge utilization and knowledge-based innovation among Malaysian researchers during the Pandemic of COVID-19*

**Proposed research framework.** Based on the comprehensive literature review, this study developed a research model that revolves around knowledge creation, knowledge acquisition, knowledge sharing, knowledge utilization, knowledge-worker productivity, and knowledge-based innovation as shown in Fig 1.

## Methodology

### Sample and data collection

The respondents of this study were knowledge workers (academic researchers) at various faculties in 9 Malaysian universities. The chosen knowledge workers were highly innovative, have full autonomy, focus on quality and quantity of research publications, keep continuity in learning, and were perceived as the important asset of their universities. In total 90 institutions, the academic researchers were around 75000 as per the universities statistics. Krejcie-morgan-table was used to arrive at 382 as a sample of this study.

### Instruments and measurement

This study has applied positive approach to collect data through questionnaire based survey. A close-ended questionnaire was the instrument used for data collection along with random sampling technique. The questionnaire was self-administered by the researcher in order to avoid any misunderstanding. This study adopted and adapted the instrument for data collection, using all three constructs (knowledge management processes, knowledge-worker

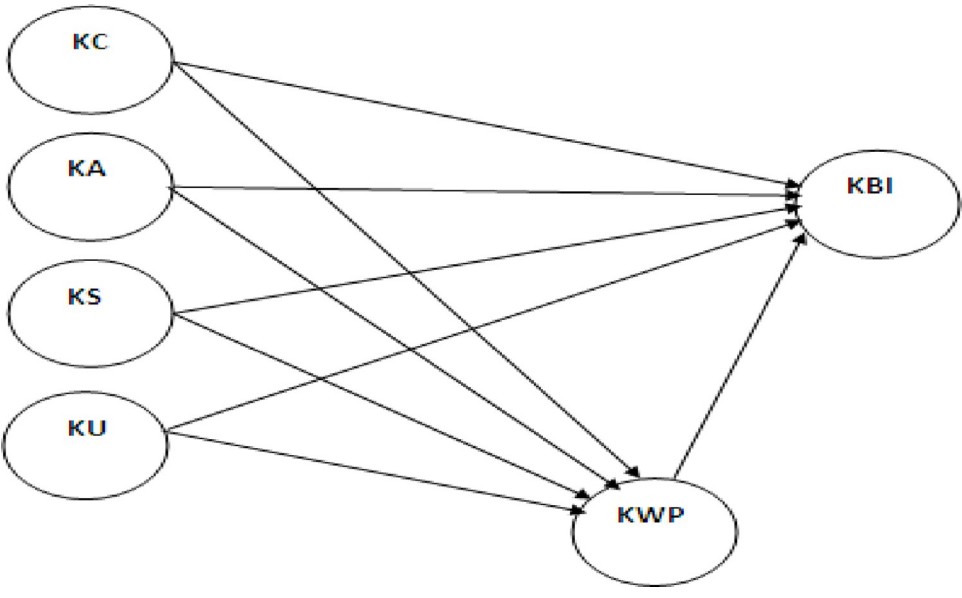

**Fig 1. Research framework.**

productivity, and knowledge-based innovation). Table 1 below shows the details of all constructs measuring items. The first construct (knowledge management processes) has four dimensions; knowledge creation, knowledge acquisition, knowledge sharing, and knowledge utilization, to measure the conception. In the same way, the second construct (knowledge worker productivity) also has three dimensions; timelines of workers, task efficiency, and autonomy, to measure the conception. The structure of constructs was arranged in accordance with guidelines utilized by earlier scholars [2, 5–7, 70]. The third construct, innovation has five dimensions; product innovation, process innovation, service innovation, radical innovation, and customer problem-solving capabilities.

## Data analysis technique

This study analyzed the collected data using smart partial least square-structure equation modeling (Smart PLS-SEM 3 version 26 software) to find results. PLS-SEM has the advantage

**Table 1. Instruments of the study.**

| Constructs | Dimensions | Number of items | Instruments |
|---|---|---|---|
| Knowledge Management Process | Knowledge creation | 9 | CEN (2004) [76] |
| | Knowledge Acquisition | 7 | Kim and Lee (2010) [77], Lia (2018) [78], Lai and Wang (2012) [79] |
| | Knowledge sharing | 9 | CEN (2004) [76] |
| | Knowledge utilization | 9 | CEN (2004) [76] |
| knowledge-worker productivity | Research autonomy at work | 3 | Morgeson and Humphrey (2006) [80] |
| | Meeting time demands | 2 | Lerner et al. (2001) [81] |
| | Work efficiency | 2 | Tangen (2005) [82] |
| Innovation | Product innovation | 3 | Wang and Ahmed (2004) [83] |
| | Problem-solving Capability | 4 | Jayachandran, Hewett, and Kaufman (2004) [84] |
| | Process Innovation | 2 | CIS (2004) [85], Asgharian (2012) [30] |
| | Service Innovation | 3 | Yen et al. (2012) [86] |
| | Radical Innovation | 3 | Shih (2018) [87] |

of offering a systematic mechanism for the validation of relationships among different constructs and allows for testing all the relationships in a single model [71, 72]. It also has the advantage of dealing with complex models [71, 72], and do not require normality of data distribution, no sample size restriction, it can accommodate nominal, ordinal, and continuous scales, and estimates multicollinearity problem [73]. The choice of PLS-SEM over other methods was due to the nature of the research problem, hypothesis, formative style of the model, and predictive nature of the study. The PLS path modeling is used in predicting relationships between latent variables [73]. In addition, software 26 has the capabilities to use the PLS-SEM technique [74]. However, the presentation of results, the method, and the technique of this study are consistent with [9, 74, 75] because of the shared similarity of styles of the model.

## Results and discussion

### Measurement model

The evaluation of the measurement model includes assessment of outer loading, composite reliability, convergent, and discriminant validity. The convergent validity of each construct can be examined via factor loading, construct reliability, and average variance extracted (AVE) [72]. Some studies have presumed that acceptable and ideal standardized value of factor loading should be in the range of 0.7 or larger than this value [88, 89], AVE estimation value should be higher than 0.50, and the composite reliability estimation value should be above 0.7 to achieve the standardized convergent validity [89–92]. Composite reliability is used to examine internal consistency and is considered better than Cronbach's alpha to estimate item loading within a casual model [91, 93]. However, items with low values were deleted and all the remaining values of composite reliability, factor loading, and AVE were greater than the threshold values in this study as shown in Table 2. Therefore, all the constructs in this study fulfilled the requirements of convergent validity.

Discriminant validity refers to the extent where the value of a latent construct is distinct from the other constructs [72], and compares the correlation among construct and the square root of AVE for that construct [94]. It is presumed that the square root of AVE of the latent construct (diagonal values) should be greater than the correlation between that construct and other constructs in the corresponding rows and columns to meet the objective of discriminant validity [94–96]. However, all the diagonal values of the square root of AVE of each construct are higher than the inter-correlation between other constructs in the model as shown in Table 3, which represents the achievement of discriminant validity in this study. This study verified the achievement of discriminant validity using the Heterotrait-Monotrait (HTMT) approach [97], and noted that all the inter-correlation between the construct of interest and the remaining constructs are lower than the 0.85 (r < HTMT0.85) threshold value as shown in Table 4.

### Structural model

In the structural model, this study applied a bootstrapping procedure (through PLS-SEM) to test the developed hypothesis. The results revealed that knowledge creation, knowledge acquisition, knowledge sharing, and knowledge utilization have positive and significant effects on knowledge-based innovation and knowledge workers' productivity among Malaysian researchers during the Pandemic of COVID-19 as shown in Table 5. Likewise, knowledge workers' productivity has positive and significant effects on knowledge-based innovation among Malaysian academic researchers during the Pandemic of COVID-19. Notably, results revealed that knowledge sharing has greater effects on knowledge-based innovation among Malaysian researchers during the Pandemic of COVID-19 as compared to knowledge creation,

**Table 2. Confirmatory factor analysis, composite reliability, and AVE values.**

| Constructs | Items | Factor Loading | Composite Reliability | AVE |
|---|---|---|---|---|
| Innovation | PRTINN1 | 0.762 | 0.861 | 0.603 |
| | PRTINN2 | 0.798 | | |
| | PRTINN3 | 0.701 | | |
| | PSCINN4 | 0.852 | | |
| | PSCINN5 | 0.861 | | |
| | PSCINN6 | 0.756 | | |
| | PRSINN7 | 0.758 | | |
| | PRSINN8 | 0.863 | | |
| | SRCINN9 | 0.748 | | |
| | SRCINN10 | 0.832 | | |
| | RDCINN11 | 0.867 | | |
| | RDCINN12 | 0.794 | | |
| Knowledge Creation | KNC1 | 0.784 | 0.885 | 0.642 |
| | KNC2 | 0.709 | | |
| | KNC3 | 0.788 | | |
| | KNC4 | 0.850 | | |
| | KNC5 | 0.763 | | |
| | KNC7 | 0.798 | | |
| | KNC8 | 0.789 | | |
| | KNC9 | 0.799 | | |
| Knowledge Acquisition | KNA1 | 0.872 | 0.895 | 0.634 |
| | KNA2 | 0.863 | | |
| | KNA3 | 0.789 | | |
| | KNA4 | 0.852 | | |
| | KNA5 | 0.795 | | |
| Knowledge Sharing | KNS1 | 0.796 | 0.865 | 0.593 |
| | KNS2 | 0.876 | | |
| | KNS3 | 0.768 | | |
| | KNS4 | 0.898 | | |
| | KNS5 | 0.769 | | |
| | KNS6 | 0.788 | | |
| | KNS7 | 0.793 | | |
| | KNS9 | 0.889 | | |
| Knowledge Utilization | KNU1 | 0.798 | 0.843 | 0.675 |
| | KNU2 | 0.832 | | |
| | KNU3 | 0.787 | | |
| | KNU4 | 0.796 | | |
| | KNU5 | 0.798 | | |
| | KNU6 | 0.795 | | |
| | KNU8 | 0.774 | | |
| | KNU9 | 0.887 | | |
| Knowledge Workers Productivity | JAW1 | 0.796 | 0.878 | 0.587 |
| | JAW2 | 0.797 | | |
| | JAW3 | 0.790 | | |
| | MTD1 | 0.783 | | |
| | MTD2 | 0.893 | | |
| | WE1 | 0.698 | | |
| | WE2 | 0.842 | | |

**Table 3. Discriminant validity.**

| Construct | Knowledge Creation | Knowledge Acquisition | Knowledge Sharing | Knowledge Utilization | Knowledge-Workers Productivity | Knowledge Based Innovation |
|---|---|---|---|---|---|---|
| Knowledge Creation | **0.875** | | | | | |
| Knowledge Acquisition | 0.476 | **0.864** | | | | |
| Knowledge Sharing | 0.297 | 0.426 | **0.899** | | | |
| Knowledge Utilization | 0.509 | 0.365 | 0.673 | **0.886** | | |
| Knowledge-Workers Productivity | 0.359 | 0.461 | 0.389 | 0.285 | **0.796** | |
| Knowledge Based Innovation | 0.408 | 0.362 | 0.354 | 0.349 | 0.496 | **0.905** |

knowledge acquisition, knowledge utilization, and knowledge workers productivity (Fig 2). Hence, knowledge sharing has better results that could enhance the level of knowledge-based innovation among Malaysian academic researchers during the Pandemic of COVID-19. In addition, knowledge worker productivity mediates the relationship between knowledge creation, knowledge acquisition, knowledge sharing, and knowledge utilization with the knowledge-based innovation among Malaysian researchers during the Pandemic of COVID-19.

The results of the study indicate that the values of Q-Square are greater than zero which shows that the path model's predictive relevance exists in this study. Therefore, on the basis of results, it can be inferred that knowledge creation, knowledge acquisition, knowledge sharing, knowledge utilization, knowledge workers productivity are the convenient sources to stimulate knowledge-based innovation among Malaysian researchers during the Pandemic of COVID-19. In the first hypothesis H1, it was proposed that Knowledge creation has positive significant effects on knowledge-based innovation among Malaysian academic researchers during the Pandemic of COVID-19. The results of the structural model show a positive and significant effect of knowledge creation on knowledge-based innovation ($\beta = 0.\,0.377$, $p < 0.05$; Table 5). In the tenth hypothesis H10, the indirect effect of knowledge creation on knowledge-based innovation through knowledge-worker productivity represents a positive and significant effect ($\beta = 0.065$, $p < 0.05$; Table 6). Therefore, on the basis of results, it can be inferred that knowledge creation in the context of self-creation and reflection, students feedback along with knowledge management processes from the perspective of knowledge-workers productivity leads toward greater innovation in the educational sector during the Pandemic of COVID-19. However, this study is in line with the findings of earlier studies [18, 20], in the case of knowledge creation and task efficiency and in line with findings of Ramezan [67] as well as in the instance of the relationship between productivity and innovation.

**Table 4. Heterotrait-Monotrait Ration (HTMT).**

| Construct | Knowledge Creation | Knowledge Acquisition | Knowledge Sharing | Knowledge Utilization | Knowledge-Workers Productivity | Knowledge Based Innovation |
|---|---|---|---|---|---|---|
| Knowledge Creation | | | | | | |
| Knowledge Acquisition | 0.472 | | | | | |
| Knowledge Sharing | 0.462 | 0.438 | | | | |
| Knowledge Utilization | 0.509 | 0.411 | 0.683 | | | |
| Knowledge-Workers Productivity | 0.259 | 0.537 | 0.469 | 0.261 | | |
| Knowledge Based Innovation | 0.378 | 0.373 | 0.464 | 0.289 | 0.296 | |

**Table 5. Results of the structural model analysis (hypothesis testing).**

| Hypothesis | Relationship | St. Beta | Sample Mean | SD | T-Value | Decision | $R^2$ | $F^2$ | VIF | $Q^2$ |
|---|---|---|---|---|---|---|---|---|---|---|
| H1 | KNC→ INN | 0.377 | 0.523 | 0.083 | 2.028 | Supported | 0.382 | 0.193 | 1.903 | 0.279 |
| H2 | KNA→ INN | 0.283 | 0.327 | 0.047 | 2.852 | Supported | | 0.147 | 1.84 | |
| H3 | KNS→ INN | 0.527 | 0.318 | 0.094 | 3.910 | Supported | | 0.160 | 1.793 | |
| H4 | KNU→ INN | 0.261 | 0.382 | 0.067 | 2.739 | Supported | | 0.205 | 1.488 | |
| H5 | KNC→ KWP | 0.392 | 0.565 | 0.073 | 2.063 | Supported | 0.473 | 0.253 | 1.983 | 0.832 |
| H6 | KNA→ KWP | 0.352 | 0.428 | 0.078 | 2.674 | Supported | | 0.198 | 1.783 | |
| H7 | KNS→ KWP | 0.473 | 0.384 | 0.089 | 3.354 | Supported | | 0.169 | 1.893 | |
| H8 | KNU→ KWP | 0.362 | 0.419 | 0.052 | 2.563 | Supported | | 0.217 | 1.893 | |
| H9 | KWP→ INN | 0.304 | 0.442 | 0.073 | 2.983 | Supported | 0.267 | 0.193 | 1.983 | 0.283 |

In the second hypothesis H2, it was proposed that knowledge acquisition has positive significant effects on the knowledge-based innovation among Malaysian researchers during the Pandemic of COVID-19. The results of the structural model show positive significant effects of knowledge acquisition on the knowledge-based innovation ($\beta = 0.283$, $p < 0.05$; Table 5). While in hypothesis eleven H11, the indirect effect of knowledge acquisition on knowledge-based innovation through knowledge-worker productivity represents positive significant effects ($\beta = 0.047$, $p < 0.05$; Table 6). In the third hypothesis H3, it was proposed that Knowledge sharing has positive significant effects on the knowledge-based innovation among Malaysian academic researchers during the Pandemic of COVID-19. The results of the structural model show a positive and significant effect of knowledge sharing on knowledge-based innovation ($\beta = 0.527$, $p < 0.05$; Table 5). While in hypothesis H12, the indirect effect of knowledge sharing on knowledge-based innovation through knowledge-worker productivity represents positive and significant effects ($\beta = 0.057$, $p < 0.05$; Table 6). However, the results of this study are consistent with [19] in the context of knowledge sharing and productivity. In the fourth hypothesis H4, it was proposed that knowledge utilization has a positive and significant

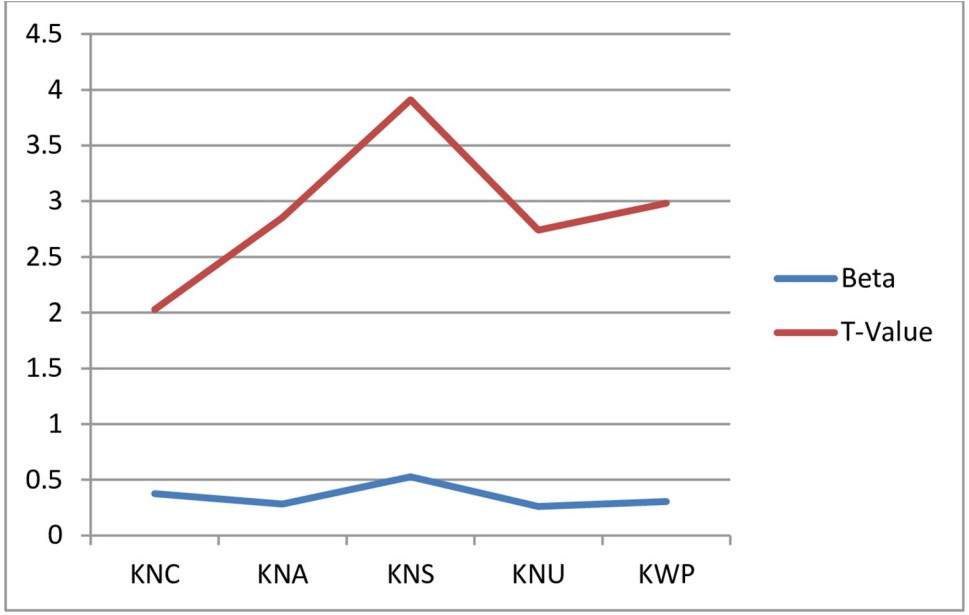

**Fig 2. Effects on innovation.**

**Table 6. Results of indirect effects (hypothesis testing).**

|  | Relationship | St. Beta | SM | SD | T-value | CILL | CIUL | Decision |
|---|---|---|---|---|---|---|---|---|
| H10 | KNC*KWP→INN | 0.065 | 0.475 | 0.084 | 2.931 | 0.052 | 0.078 | Supported |
| H11 | KNA*KWP→INN | 0.047 | 0.481 | 0.063 | 2.838 | 0.048 | 0.069 | Supported |
| H12 | KNS*KWP→INN | 0.057 | 0.481 | 0.298 | 2.786 | 0.074 | 0.149 | Supported |
| H13 | KNU*KWP→INN | 0.036 | 0.483 | 0.272 | 2.362 | 0.037 | 0.142 | Supported |

CILL–Confidence interval lower level, CIUL–Confidence interval upper level, $p < 0.05$.

relationship with knowledge-based innovation among Malaysian academic researchers during the Pandemic of COVID-19. The results of the structural model show a positive and significant effect of knowledge utilization on knowledge-based innovation ($\beta = 0.261$, $p < 0.05$; Table 5). While in hypothesis H13, the indirect effect of knowledge utilization on knowledge-based innovation through knowledge-worker productivity represents positive significant effects ($\beta = 0.036$, $p < 0.05$; Table 6). In the fifth hypothesis, knowledge creation has a positive significant relationship ($\beta = 0.392$, $p < 0.05$; Table 5) with the Malaysian academic researcher's productivity during the Pandemic of COVID-19. In the sixth hypothesis, knowledge acquisition has a positive significant relationship ($\beta = 0.352$, $p < 0.05$; Table 5) with the Malaysian academic researcher's productivity during the Pandemic of COVID-19. Similarly, in the seventh hypothesis, knowledge sharing has a positive significant relationship with the ($\beta = 0.473$, $p < 0.05$; Table 5) with the Malaysian academic researcher's productivity during the Pandemic of COVID-19.

Likewise, in the eighth hypothesis, knowledge utilization has a positive and significant relationship with the ($\beta = 0.362$, $p < 0.05$; Table 5) Malaysian academic researchers' productivity during the Pandemic of COVID-19. Furthermore, in the ninth hypothesis, the results have shown a positive and significant effect of knowledge workers' productivity ($\beta = 0.\ 0.304$, $p < 0.05$; Table 5) on the knowledge-based innovation among Malaysian academic researchers during the Pandemic of COVID-19.

Notably, the findings of this research indicated that some theoretical contributions have been made. Firstly, this study has integrated the dimensions of the knowledge management process and merged the literature based on the findings from emerging economies during the Pandemic of COVID-19. Such integration allows for holistic views of the proposed research model that visualizes and validates all influential factors on knowledge-based innovation. Thus, results have confirmed that each factor significantly contributes to the knowledge-based innovation among Malaysian academic researchers during the Pandemic of COVID-19. The results have provided a theoretical foundation for further research and a detailed analysis of the influential factors. The results have provided a bigger picture for a better understanding that the knowledge management process and knowledge workers' productivity are the crucial factors in knowledge-based innovation among researchers during the Pandemic of COVID-19.

Significantly, the findings imply that educational institutions can focus their efforts on the knowledge management process and knowledge workers' productivity to enhance knowledge-based innovation among researchers during the Pandemic of COVID-19. The motivation towards knowledge creation, acquisition, sharing sessions, and knowledge utilization strategies would be helpful in innovation consequence, as regular knowledge sharing sessions, knowledge workers productivity can motivate beginners towards high innovation. By developing unique knowledge management strategies that better suit researchers needs can lead universities towards high innovation and ultimately towards high ranking. High knowledge-based

innovation can lead the universities to be perceived as market leaders and top brands among communities.

In addition, the results of this study also contribute to the development of the real-world application of knowledge-based innovation among researchers in Malaysian universities in several ways. Firstly, this study found that the greater the knowledge sharing, the higher level of knowledge-based innovation will occur during the Pandemic of COVID-19. Therefore, universities are required to cultivate researchers' anticipated usefulness and provide researchers with well-deserved acknowledgment and encouragement towards outstanding knowledge-based innovation. In this regard, rewards, funding, access to digital libraries and software, and facilities can be given to the researchers for knowledge-based innovation such as efficiently solving work-related issues. Secondly, the motivating knowledge-sharing sessions can provide ideas on how the universities can promote knowledge-based innovation culture among researchers to gain a competitive advantage. This culture can guide the universities management on how to gain its market competitive advantage, and allow for the creation, acquisition, sharing, and utilization of new knowledge to improve work policies, and enhance work practices. Thirdly, the universities should create awareness of crucial knowledge creation, acquisition, sharing, and utilization drivers among researchers that are highly associated with knowledge-based innovation. Fourthly, all the factors positively and significantly affect the knowledge-based innovation which indicates that researchers in universities can be motivated through shared collaboration that eventually leads to gaining competitive advantage and better research productivity. In this study, knowledge sharing is seen to have a greater influence on knowledge-based innovation among researchers. Therefore, to enhance the knowledge-sharing culture among researchers, the universities should hold regular sessions by inviting experts and trainers to engage other staff members. Furthermore, developing digital blogs and online meetings can create virtual communities for easy knowledge sharing among researchers during the Pandemic of COVID-19.

## Conclusion

This study aims to evaluate the effects of the knowledge management process on knowledge-based innovation along with mediating role of academic researchers' productivity during the Pandemic of COVID-19 at Malaysia. Data was collected from the academic researchers in nine Malaysian universities during the Pandemic of COVID-19. The study has selected a sample of 382 but only 304 respondents participated in this study. The results confirmed that knowledge creation, knowledge acquisition, knowledge sharing, knowledge utilization, and knowledge workers' productivity have positive and significant effects on the knowledge-based innovation among academic researchers in Malaysian universities during the Pandemic of COVID-19. In addition, knowledge creation, knowledge acquisition, knowledge sharing, and knowledge utilization have a positive and significant relationship with the Malaysian academic researcher's productivity during the Pandemic of COVID-19. The results of the study have confirmed that knowledge-workers productivity mediates the relationship between knowledge management processes (knowledge creation, knowledge acquisition, knowledge sharing, and knowledge utilization) and knowledge-based innovation among academic researchers in Malaysian universities during the Pandemic of COVID-19. Nevertheless, the results of this study have verified Drucker's knowledge-workers productivity theory owing to the mediating role of knowledge-worker productivity, and have also verified the knowledge-based view of the firm theory in the setting of Pandemic COVID-19. However, while interpreting the results of this study, the readers should know about the scope and limitations of collected data. To further enrich understanding in the analysis, future studies can examine the inter-correlation

between the constructs of the knowledge management process and the moderating role of knowledge workers' productivity. In addition, this study was only limited to Drucker's knowledge-worker productivity theory and knowledge-based view of the firm theory, as such future studies can assess the application of Bloom's Taxonomy concept to further enrich the body of knowledge.

## Supporting information

**S1 Appendix.**
(DOCX)

## Author Contributions

**Conceptualization:** Fazal ur Rehman.

**Data curation:** Fazal ur Rehman.

**Formal analysis:** Fazal ur Rehman.

**Investigation:** Fazal ur Rehman.

**Methodology:** Fazal ur Rehman, Muhammad Saeed Shahbaz.

**Project administration:** Fazal ur Rehman.

**Resources:** Fazal ur Rehman, Basheer M. Al Ghazali.

**Software:** Fazal ur Rehman.

**Supervision:** Fazal ur Rehman, Hishamuddin Ismail.

**Validation:** Fazal ur Rehman, Muhammad Mujtaba Asad.

**Visualization:** Fazal ur Rehman.

**Writing – original draft:** Fazal ur Rehman.

**Writing – review & editing:** Fazal ur Rehman, Ali Zeb.

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
