## [Decision Letter · Decision Letter 0]

3 Nov 2021

PONE-D-21-31960Knowledge Management Process, Knowledge based Innovation: Does Academic Researchers Productivity mediate during the Pandemic of COVID-19.?PLOS ONE

Dear Dr. Fazal Ur Rehman,

Thank you for submitting your manuscript to PLOS ONE. After careful consideration, we feel that it has merit but does not fully meet PLOS ONE’s publication criteria as it currently stands. Therefore, we invite you to submit a revised version of the manuscript that addresses the points raised during the review process.

We look forward to receiving your revised manuscript.

Kind regards,

Rogis Baker, Ph.D

Academic Editor

PLOS ONE

Journal Requirements:

2. Please include additional information regarding the survey or questionnaire used in the study and ensure that you have provided sufficient details that others could replicate the analyses. For instance, if you developed a questionnaire as part of this study and it is not under a copyright more restrictive than CC-BY, please include a copy, in both the original language and English, as Supporting Information

4. Please ensure that you refer to Figure 2 in your text as, if accepted, production will need this reference to link the reader to the figure.

5. Please upload a copy of Figure 3 to which you refer in your text on page 30. If the figure is no longer to be included as part of the submission please remove all reference to it within the text

6. We note you have included a table to which you do not refer in the text of your manuscript. Please ensure that you refer to Tables 2, 3 and 4 in your text; if accepted, production will need this reference to link the reader to the Table.

Reviewers' comments:

Reviewer's Responses to Questions

**Comments to the Author**

1. Is the manuscript technically sound, and do the data support the conclusions?

Reviewer #1: Yes

Reviewer #2: Partly

Reviewer #3: Yes

2. Has the statistical analysis been performed appropriately and rigorously? 

Reviewer #1: Yes

Reviewer #2: I Don't Know

Reviewer #3: Yes

3. Have the authors made all data underlying the findings in their manuscript fully available?

Reviewer #1: Yes

Reviewer #2: No

Reviewer #3: Yes

4. Is the manuscript presented in an intelligible fashion and written in standard English?

Reviewer #1: Yes

Reviewer #2: No

Reviewer #3: Yes

5. Review Comments to the Author

Reviewer #1: (1) The study design is not explicitly stated. There are suggestions of what the study design the authors used, however, it needs to be stated at the start of the methodology section.

(2) The population (i.e. number of academic staff and number of Malaysian universities). The authors mention that 389 academic staff were interviewed. Please state the population size.

Reviewer #2: First, the rationale for the study being linked to the COVID-19 pandemic is not readily apparent. There is no attempt to compare this study with a time when there was no pandemic, there appears no relevance linked to knowledge products produced on the pandemic - there is simply no indication of the relevance of the pandemic except for the study having happened at the same time as the pandemic.

Second, the article was repetitive, at times incomprehensible and although there was a proof-reading certificate attached, was not written in standard English.

Third and most importantly - there are some serious methodological concerns. For example, there is little information given about the sampling - the approach or methodology - to assess whether this was a rigorous approach. In fact some valuable information was provided but as an after thought in the conclusion section. More importantly however is the fact that the data was collected via a self-assessment tool. However, the hypotheses argue for a positive and significant relationship between KM processes and innovation and productivity when the study respondents would have clear self-interest in portraying their innovation and productivity in a positive light. Therefore their is an inherent bias in the respondents answers which isn't acknowledged at all. At the very least the study should be seen as a one evaluating the positive relationship based on a self-assessment but this limitation isn't readily referenced or discussed when looking at the findings and drawing conclusions.

In addition, the logic and soundness of the analysis framework is not set out, unless it is intended to be the list of definitions of KM products, processes, innovation products etc in the literature review.

finally, this leads me to question what the relevance of the conclusions are - if it was to validate the model, for what purpose? this is not set out at all. Particularly when so much is made of the fact that the model was assessed during the COVID-19 pandemic - what is the relevance for the model and why did it need to be assessed? Also, Druker's theory and model has been validated before - what is the need for it to be validated in academic settings in Malaysia during the global pandemic? All of these points are not addressed. As a reader I have to admit my first and foremost thought was 'why'?

Reviewer #3: This is a very detailed and well-written paper. The use of this model during the pandemic was unique and interesting. The authors are also well-steeped in the KM literature as evidence by the theories and research cited. I would recommend including a bit more in the conclusion about how others outside of Malaysia may be able to use these results, despite the limitation (i.e., research only conducted within Malaysia).

6. PLOS authors have the option to publish the peer review history of their article (what does this mean?). If published, this will include your full peer review and any attached files.

Reviewer #1: **Yes: **JAMES KARIUKI NGUMO

Reviewer #2: No

Reviewer #3: No

---

## [Author Response · Author response to Decision Letter 0]

22 Nov 2021

Dear Reviewers and Editors

Thank you very much for the review of our manuscript, your time, and providing interesting guidelines, and comments on our paper…..we tried our best to address all the issues in our manuscript as per your respected suggestions. 

Thank You

---

## [Editor Report · Decision Letter 1]

6 Dec 2021

Knowledge Management Process, Knowledge based Innovation: Does Academic Researchers Productivity mediate during the Pandemic of COVID-19.?

PONE-D-21-31960R1

Dear Dr. FAZAL UR REHMAN,

We’re pleased to inform you that your manuscript has been judged scientifically suitable for publication and will be formally accepted for publication once it meets all outstanding technical requirements.

Kind regards,

Rogis Baker, Ph.D

Academic Editor

PLOS ONE
---

## [Editor Report · Acceptance letter]

13 Dec 2021

PONE-D-21-31960R1 

Knowledge management process, knowledge based innovation: does academic researcher’s productivity mediate during the pandemic of covid-19.? 

Dear Dr. REHMAN:

I'm pleased to inform you that your manuscript has been deemed suitable for publication in PLOS ONE. Congratulations! Your manuscript is now with our production department. 

Kind regards, 

on behalf of

Dr. Rogis Baker 

Academic Editor

PLOS ONE